# TRAINING FEATURE ATTRIBUTION FOR VISION MODELS

## ABSTRACT

Deep neural networks are often considered opaque systems, prompting the need for explainability methods to improve trust and accountability. Existing approaches typically attribute test-time predictions either to input features (e.g., pixels in an image) or to influential training examples. We argue that both perspectives should be studied jointly. This work explores *training feature attribution*, which links test predictions to specific regions of specific training images and thereby provides new insights into the inner workings of deep models. Our experiments on vision datasets show that training feature attribution yields fine-grained, test-specific explanations: it identifies harmful examples that drive misclassifications and reveals spurious correlations, such as patch-based shortcuts, that conventional attribution methods fail to expose.

## 1 INTRODUCTION

Deep neural networks have achieved state-of-the-art performance across a wide range of domains, including image recognition, natural language processing, and multimodal reasoning (He et al., 2016; Devlin et al., 2019; Radford et al., 2021). However, this impressive performance comes at the cost of transparency: modern deep models operate as complex, highly-parameterized *black boxes*, where the reasoning behind individual predictions is often opaque (Lipton, 2018). This opacity can undermine user trust, hinder debugging, and conceal harmful biases or spurious correlations (Arjovsky et al., 2019; DeGrave et al., 2021). In high-stakes applications such as healthcare, finance, or autonomous systems, understanding *why* a model makes a specific decision is as important as the decision's accuracy itself (Rudin, 2019; Doshi-Velez & Kim, 2017). Explainability methods aim to bridge this gap by attributing model predictions to interpretable causes, enabling practitioners to verify alignment with domain knowledge, detect potential failures, and ensure accountability. This is also a way to improve interaction between humans and AI systems (Wickramasinghe et al., 2020).

Within the literature on eXplainable AI (XAI), two main *attribution* paradigms can be distinguished: feature attribution (FA), which highlights the parts of a test input most responsible for its prediction (e.g., pixels in image classification), and training data attribution (TDA), which identifies the training examples most influential for a given test prediction.

While both provide valuable insights, each has inherent limitations. Feature attribution ignores *where* in the training data the model learned its decisive features, while TDA ignores *what* aspects of those examples matter most (see Figure 1). For instance, a feature attribution map might highlight a "striped" region in a zebra image without indicating whether the stripes were learned from zebras or from unrelated patterns in the training set; conversely, TDA might flag a specific training image without clarifying which region of it was influential.

This gap motivates **training feature attribution** (TFA), a framework that connects test predictions to specific regions of training examples. By combining TDA with FA, such an approach enables us to answer the question, *Which parts of which training images are most responsible for the model's decision on this test image?*

**Related works**   While feature attribution and training data attribution have each been extensively studied in isolation, there has been comparatively little research on integrating these approaches into a unified framework. A notable exception is the exploration of training feature attribution

$$f\left(\hat{\theta}, x_{\text{test}}\right) \quad \underbrace{- \quad x_{\text{test}} \quad -}_{} \quad \text{test example}$$

$$\mathcal{A}\left(\left\{\begin{array}{ccc} - & x_1 & - \\ & \cdots & \\ - & x_n & - \end{array}\right\}\right) \quad \text{training data}$$

prediction on a test example — $f\left(\hat{\theta}, x_{\text{test}}\right)$

training algorithm — $\mathcal{A}$

Figure 1: A prediction on a test example depends both on the features of the example, as well as on features learned from the training examples through the trained parameters.

within natural language processing (NLP) for artifact discovery (Han et al., 2020; Pezeshkpour et al., 2021), as well as token-wise influence functions in large language models (Grosse et al., 2023). We build upon these efforts by extending the framework to vision data, where the notion of "features" is inherently less well-defined. Unlike tokens in NLP, which typically carry semantic meaning individually, pixels in images convey limited information in isolation and gain significance primarily through their spatial interactions with other pixels.

For vision tasks, other related efforts include concept-based attribution methods, which decompose model activations into human-interpretable concepts (Kim et al., 2018); prototype-based explanations, such as ProtoPNet (Chen et al., 2019), which connect predictions to similar image regions; and Visual-TCAV (De Santis et al., 2024), which integrates the TCAV framework (Kim et al., 2018) with saliency maps for predefined concepts. In a different context, the concept of computing pixel-wise influence can be traced back to the seminal work of Koh & Liang (2017), who first applied classical influence functions to deep learning. In their work, it was used as a way to create data poisoning; however, its potential as an explainability tool was not recognized.

**Contributions**

- We introduce training feature attribution of vision models, and propose a practical algorithm for estimating saliency maps (Section 3). This algorithm is quantitatively validated by masking training images and retraining (Section 4.3);

- We introduce a simplified analytical setup where the TFA method correctly recovers the important training feature for a given test example (Section 3.1);

- We present 2 practical use cases where TFA is more insightful for debugging trained deep neural networks than using only TDA or FA (Section 5).

## 2 BACKGROUND : ATTRIBUTING PREDICTIONS TO EITHER FEATURES OR TRAINING DATA

### 2.1 TRAINING DATA ATTRIBUTION

Example-based explanation methods (surveyed in Poché et al., 2023) offer a natural way to interpret machine learning models, where explanations are conveyed through representative samples rather than abstract feature scores. This paradigm aligns with human reasoning, as people often justify decisions by referring to familiar cases: "This looks like something I've seen before." It reflects cognitive processes in which new observations are understood by comparison with previously encountered examples, allowing concepts to be formed through such comparisons (Miller, 2019; Byrne, 2016; Gentner, 1983).

Within this family, various strategies exist: prototype methods select representative instances from the training data (Chen et al., 2019); concept-based methods (Kim et al., 2018; Fel et al., 2023) explain predictions in terms of higher-level semantic factors; and criticisms, or irregular instances, highlight unusual or atypical cases in the data (Kim et al., 2016).

Another form of example-based explanation is *training data attribution* (TDA), which aims to trace a model's prediction back to the *training* examples that most influenced it. Each training instance is assigned an importance score reflecting its effect on the model's behavior for a specific test case. A positive score indicates that the example supported the prediction by pushing it toward the correct label, while a negative score means it opposed the prediction, pulling it toward an incorrect outcome.

TDA approaches vary in how they estimate influence. Influence function-based methods (Koh & Liang, 2017) compute the hypothetical effect of upweighting or removing an example at training

time. Approaches based on gradient or representation similarity estimate influence by comparing the model's response to training and test inputs (Charpiat et al., 2019; Hanawa et al., 2021; Pruthi et al., 2020). Game-theoretic frameworks such as DataShap (Ghorbani & Zou, 2019) approximate Shapley values to assign each training point a contribution score.

**Mathematical Formulation**  Consider a supervised classification problem with a training set $\mathcal{D}_{\text{train}} = \{z_i^{\text{train}}\}_{i=1}^{N}$ and a test set $\mathcal{D}_{\text{test}} = \{z_j^{\text{test}}\}_{j=1}^{M}$, where $z = (x, y)$ denotes an input-label pair. A trained model $f_\theta$ is obtained by searching parameters $\hat{\theta}$ that minimize the empirical risk:

$$\hat{\theta} = \underset{\theta}{\arg\min} \ \frac{1}{N} \sum_{i=1}^{N} \ell(f_\theta(x_i^{\text{train}}), y_i^{\text{train}})$$

where $\ell$ is the loss function. At the end of training, the predictor $f_{\hat{\theta}}$ is therefore influenced by all examples $z_i^{\text{train}}$ seen during training.

A *training data attribution* method assigns an importance score $S(z_i^{\text{train}}, z_j^{\text{test}})$, measuring the effect of including a particular training point $z_i^{\text{train}}$ on test predictions of the trained model $f_{\hat{\theta}}(x_j^{\text{test}})$ or their losses. These scores can be positive for training points that support the test label $y_j^{\text{test}}$, or negative for training examples that oppose the current label, which for instance happens when a training example is considered similar to the test example by the model, but labeled as a different class. Ranking training points by their TDA score $S(\cdot, z_j^{\text{test}})$ provides insight into which training instances most influenced (both positively and negatively) the model's decision for a given test example.

## 2.2 Feature attribution

Feature attribution (FA) methods aim to explain a model's prediction for a given test input by assigning an importance score to each input feature (e.g., pixels in an image). Unlike data attribution, which seeks to identify influential training examples, feature attribution answers: Which parts of the input were most responsible for this prediction? This is especially useful for extracting more interpretable rules from trained deep neural networks, where the gigantic number of individual parameters renders the behavior of the model difficult to interpret.

Early approaches to feature attribution include the deconvolutional network method and occlusion sensitivity analysis (Zeiler & Fergus, 2014), as well as simple gradient-based saliency maps that highlight input regions most relevant to a class prediction (Simonyan et al., 2013). More recent methods fall into two broad families: perturbation-based and gradient-based. Perturbation-based methods, such as LIME (Ribeiro et al., 2016) and SHAP (Lundberg & Lee, 2017), measure the effect of systematically masking or altering input features to estimate their importance. LIME approximates the model locally with an interpretable surrogate, while SHAP employs Shapley values from cooperative game theory to attribute contributions to each feature.

Gradient-based methods instead rely on the derivatives of the model's output with respect to its input. Examples include Integrated Gradients (Sundararajan et al., 2017), which accumulates gradients along a path from a baseline to the input, and SmoothGrad (Smilkov et al., 2017), which averages gradients computed on noisy versions of the input to improve robustness. For convolutional networks (CNNs) applied to vision tasks, techniques such as Grad-CAM (Selvaraju et al., 2017) and Grad-CAM++ (Chattopadhay et al., 2018) generate visual explanations by highlighting the spatial regions of an image most influential for a specific class prediction.

Despite their popularity, recent work shows that saliency methods can be misleading: they may produce similar maps even after randomly resampling model parameters or permuting labels, passing visual 'sanity checks' while not reflecting what the model actually learned (Adebayo et al., 2018).

## 3 Approach: attribution of test-time predictions to features seen during training

From a learning-theoretic perspective, the features used at test time in trained deep networks are learned entirely from the training set (Figure 1). A model cannot reliably assign importance to a feature it has never observed during training; for example, if most cow images in the training data

show grassy fields, the model may misclassify a cow in a desert as a camel, even if the cow is clearly visible to a human observer (Arjovsky et al., 2019; Beery et al., 2018). As an ideal long term goal, we would like to have an explainability tool able to surface these implicit mechanisms in terms of high level features.

While both FA and TDA offer valuable insights, neither is complete on its own: feature attribution ignores where the model learned those features from, while training data attribution does not reveal *which parts* of the training examples were most important. Our aim is to combine the strengths of both approaches, creating *training feature attribution* methods that connect test-time predictions back to specific regions of specific training examples.

### 3.1 ANALYTIC TOY EXAMPLE

To make motivation more concrete, we analyze a simple linear ridge regression model in $\mathbb{R}^2$ amenable to full analytical treatment (detailed derivation is provided in Appendix A). Define $y = x_1 + x_2$ to be the ground truth rule to learn. We are given a training dataset $\{(x_i, y_i)\}_{i \leq n}$ where for $i \in \{1, ..., n-1\}$, examples $x_i = (x_{i1}, 0)$ lie on the canonical $e_1$ axis, while a single informative point reveals the signal in the $e_2$ direction $x_n = (0, c)$ with $c \neq 0$.

**TDA** In the closed-form solution to ridge regression, we can compute the exact contribution of each training point to the learned function using the representer decomposition:

$$f_{\hat{\theta}}(x_*) = \sum_{i=1}^n \alpha_i y_i, \qquad \alpha_i = x_*^\top (X^\top X + \lambda I)^{-1} x_i.$$

For a test point $x_* = (0, t)$, $t \neq 0$, we obtain $\alpha_i = \frac{t}{c^2 + \lambda} x_{i2}$, which gives $\alpha_i = 0$ for $i \neq n$, and $\alpha_n y_n = \frac{tc^2}{c^2 + \lambda} = f_{\hat{\theta}}(x_*)$. TDA correctly assigns the entire prediction to the single informative training example $(x_n, y_n)$.

**TFA** We can decompose this effect even further down to the contribution of individual features in $\alpha_i$ coefficients. Let $A = (X^\top X + \lambda I)^{-1}$. Then

$$f_{\hat{\theta}}(x_*) = \sum_{i=1}^n y_i \sum_{k=1}^2 \beta_{i,k} \qquad \beta_{i,k} = x_{ik} \, (e_k^\top A x_*)$$

For our test example $x_* = (0, t)$, $e_1^\top A x_* = 0$, hence $\beta_{i,1} = 0$ for all $i$. Meanwhile, $\beta_{i,2} = \frac{t}{c^2 + \lambda} x_{i2}$, thus $\beta_{i,2} = 0$ for all $i \neq n$ and $y_n \beta_{n,2} = \frac{tc^2}{c^2 + \lambda} = f_{\hat{\theta}}(x_*)$. TFA would here show that only the $x_{n,2}$ feature of that training example contributes to the prediction for example $x_*$, while all other features are irrelevant.

This illustrates how TFA refines example-based explanations by identifying not only *which training example* matters (as would TDA already do), but also *which feature within it*. In the following sections, we aim to design methods to produce similar TFA scores, at the scale of actual deep vision networks.

### 3.2 TFA FRAMEWORK: ATTRIBUTING TDA SCORES TO INPUT FEATURES

Let $S(z_i^{\text{train}}, z_j^{\text{test}})$ denote a training data attribution method, which quantifies the influence of a training point $z_i^{\text{train}}$ on the prediction for a test point $z_j^{\text{test}}$. Let $x_i^{\text{train}}$ be the training image of $z_i^{\text{train}} = (x_i^{\text{train}}, y_i^{\text{train}})$. Feature attribution methods generally attribute a given scalar prediction (e.g., the probability or logit of the predicted class) to specific features from the input of the model. The Training Feature Attribution (TFA) approach is to apply feature attribution to the scalar TDA score instead, thus identifying which regions of the input image $x_i^{\text{train}}$ are most responsible for the TDA method to deem a training example helpful or harmful. In order to obtain a practical algorithm, we need to choose a pair of TDA and FA methods.

### 3.2.1 CHOICE OF TDA METHOD: GRAD-COS

As a choice of TDA method, we select *gradient cosine similarity* (grad-cos, Charpiat et al., 2019) as the TDA score, because i) it is computationally efficient compared to influence function based methods that require inverting high dimensional Hessian matrices and ii) more importantly, as shown by Hanawa et al. (2021), grad-cos is the training data attribution method among those evaluated that best satisfies all 3 minimal requirements for similarity-based explanations (the *model randomization test* (Adebayo et al., 2018), the *identical class test*, and the *identical subclass test*), ensuring that the most influential examples it identifies are also meaningful from a human perspective. As an alternative, we also performed experiments with influence functions (Appendix D).

**Mathematical Formulation of Grad-Cos Attribution** Following Charpiat et al. (2019), suppose that we want to quantify how a small parameter update that reduces the loss on a training example $z_i^{\text{train}}$ affects the loss on a test example $z_j^{\text{test}}$. Consider a first-order Taylor expansion:

$$\mathcal{L}(z_i^{\text{train}}; \hat{\theta} + \delta\theta) \approx \mathcal{L}(z_i^{\text{train}}; \hat{\theta}) + \nabla_\theta \mathcal{L}(z_i^{\text{train}}; \hat{\theta})^\top \delta\theta$$

The reduction of the loss at $z_i^{\text{train}}$ by a small amount $\varepsilon$ can be achieved by choosing:

$$\delta\theta = -\varepsilon \frac{\nabla_\theta \mathcal{L}(z_i^{\text{train}}; \hat{\theta})}{\left\|\nabla_\theta \mathcal{L}(z_i^{\text{train}}; \hat{\theta})\right\|^2}$$

This induces a change in the loss for the test point:

$$\mathcal{L}(z_j^{\text{test}}; \hat{\theta} + \delta\theta) \approx \mathcal{L}(z_j^{\text{test}}; \hat{\theta}) + \nabla_\theta \mathcal{L}(z_j^{\text{test}}; \hat{\theta})^\top \delta\theta$$

$$= \mathcal{L}(z_j^{\text{test}}; \hat{\theta}) - \varepsilon \frac{\nabla_\theta \mathcal{L}(z_j^{\text{test}}; \hat{\theta})^\top \nabla_\theta \mathcal{L}(z_i^{\text{train}}; \hat{\theta})}{\left\|\nabla_\theta \mathcal{L}(z_i^{\text{train}}; \hat{\theta})\right\|^2}$$

which quantifies the effect of the training example $z_i^{\text{train}}$ on the loss at $z_j^{\text{test}}$. The sign of this effect indicates whether the example is *helpful* (reducing the test loss) or *harmful* (increasing the test loss).

Alternatively, a symmetric cosine-similarity version (Charpiat et al., 2019) is defined as:

$$S_{GC}(z_i^{\text{train}}, z_j^{\text{test}}) = \frac{\nabla_\theta \mathcal{L}(z_j^{\text{test}}; \hat{\theta})}{\left\|\nabla_\theta \mathcal{L}(z_j^{\text{test}}; \hat{\theta})\right\|} \cdot \frac{\nabla_\theta \mathcal{L}(z_i^{\text{train}}; \hat{\theta})}{\left\|\nabla_\theta \mathcal{L}(z_i^{\text{train}}; \hat{\theta})\right\|} = \cos\left(\nabla_\theta \mathcal{L}(z_j^{\text{test}}; \hat{\theta}), \nabla_\theta \mathcal{L}(z_i^{\text{train}}; \hat{\theta})\right) \quad (1)$$

### 3.2.2 CHOICE OF FA METHOD: GRADIENT-BASED IMPORTANCE

In the following, we focus on gradient-based feature attribution methods, which are computationally more efficient than perturbation-based methods and produce sensible saliency maps (Boggust et al., 2023; Smilkov et al., 2017; Adebayo et al., 2018). To derive a pixelwise influence map from $S_{GC}$, we analyze how small perturbations to individual pixels of the training image affect the attribution score.

Remark that $S_{GC}(\cdot, z_j^{\text{test}})$ is differentiable with respect to $x_i^{\text{train}}$ as the underlying neural network models and standard loss functions are differentiable with respect to their inputs almost everywhere[1]. Consider a small perturbation $\delta \in \mathbb{R}^d$ applied to $x_i^{\text{train}}$. A first-order Taylor expansion gives[2]:

$$S_{GC}(x_i^{\text{train}} + \delta, z_j^{\text{test}}) \approx S_{GC}(x_i^{\text{train}}, z_j^{\text{test}}) + \delta^\top \nabla_{x_i^{\text{train}}} S_{GC}(x_i^{\text{train}}, z_j^{\text{test}})$$

where $\nabla_{x_i^{\text{train}}} S_{GC}(x_i^{\text{train}}, z_j^{\text{test}})$ is the gradient of the attribution score with respect to the training image. This gradient assigns an importance score to each pixel, indicating how sensitive the attribution score is to small changes at that location, which we use as saliency map:

$$\text{Saliency} := \nabla_{x_i^{\text{train}}} S_{GC}(x_i^{\text{train}}, z_j^{\text{test}}) \quad (2)$$

We can further render these heatmaps more visually appealing by additionally applying SmoothGrad (Smilkov et al., 2017) to the saliency map (details in Appendix E).

---

[1]In practice, non-differentiable points (e.g., ReLU at zero) form a set of measure zero.
[2]For notational simplicity, we write $S_{GC}(x_i^{\text{train}}, z_j^{\text{test}})$ to mean $S_{GC}((x_i^{\text{train}}, y_i^{\text{train}}), z_j^{\text{test}})$.

## 4 EXPERIMENTS

The code for reproducing experiments (Sections 4 and 5) is available at
https://anonymous.4open.science/r/tfa-convnets-F245.

### 4.1 PIXELWISE INFLUENCE ATTRIBUTION

We use the Pascal VOC 2012 dataset (Everingham et al., 2015), which contains images from 20 object categories, including vehicles, household items, animals, and other common objects. Images may contain multiple objects, so both single-label and multi-label settings are evaluated. Notably, objects are not always centered, making the dataset well-suited for feature visualization. All images are resized to $224 \times 224$ pixels, and we use a ResNet-18 (He et al., 2016) model pretrained on ImageNet (Deng et al., 2009) for the experiments.

To isolate the effect of our method on a single semantic concept without confounding from multiple object classes, we first restrict the analysis to images containing exactly one annotated object category. We fine-tune a ResNet-18 pretrained on ImageNet for 5 epochs using the Adam optimizer (Kingma & Ba, 2015) (learning rate $10^{-4}$, batch size 32). The network is trained with a cross-entropy loss for this single-label setting. To reduce noise in the resulting heatmaps, we apply SmoothGrad (Smilkov et al., 2017, see Equation 3 in Appendix E), adding Gaussian noise with a standard deviation equal to 10% of the normalized pixel range to the input and averaging the attribution maps over $n = 50$ noisy copies of each image.

Figure 2 displays examples of resulting maps that highlight regions of the training image that are correctly identified as containing the object in the test image. In addition, we performed a series of experiments to assess the role of individual layers of a given deep architecture in Appendix F.1, and the different saliency maps obtained using different models such as vision transformers in Appendix F.2.

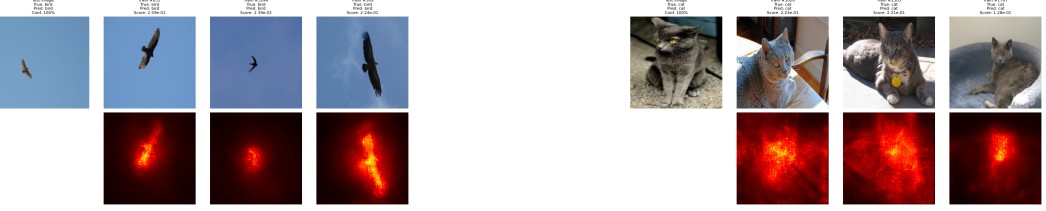

Figure 2: TFA saliencies (Equation 2) for the top-3 most influential training images per test image. Each panel (left to right): test image, influential training images, and their influence maps (smoothed using Equation 3).

### 4.2 DEPENDENCE ON THE TEST IMAGE

A key property of our method is that influence maps are *test-specific*: the same training image can produce different saliency patterns depending on the test instance, providing more specific explanations than using either feature-level attribution or training data attribution alone. To illustrate this effect, we consider the multi-label setting and select two test images from different classes (e.g., *person* and *dog*). We then compute pixelwise influence scores for the *same* training image containing both classes. As shown in Figure 3, the resulting heatmaps differ: the *person* region of the training image is most influential for the test image labeled "person," whereas the *dog* region is most influential for the test image labeled "dog."

### 4.3 QUANTITATIVE EVALUATION

We quantitatively test whether pixelwise influence maps obtained using Grad-Cos + SmoothGrad TFA identify the training pixels that most affect a given test example. On CIFAR-10 (Krizhevsky, 2009), we train a lightweight CNN for 10 epochs (Adam, lr $= 10^{-3}$, batch size 64) on 90% of the training set and keep the remaining 10% as a holdout pool. We then randomly pick 50 test images $x^{\text{test}}$ and, for each, select the $M{=}20$ most positively influential holdout images using Grad-Cos on

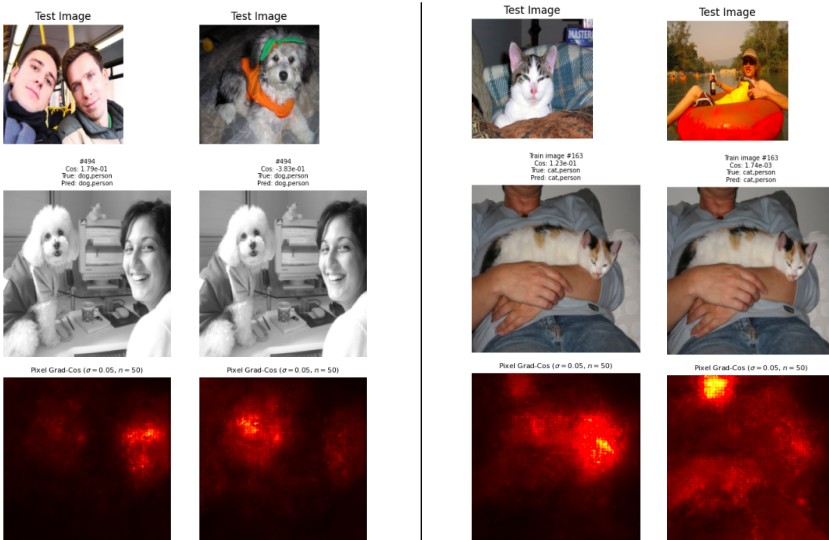

Figure 3: Two examples showing test-image dependence of pixelwise influence maps. In each example, the same training image yields different saliency patterns depending on the test image label. Left pair: *dog* vs *person* ; Right pair: *cat* vs *person*.

parameter gradients. For each selected pair $(x^{\text{train}}, x^{\text{test}})$, we compute a smoothed pixelwise influence map (SmoothGrad with Gaussian noise $\sigma=0.05$ of the normalized range, $n=30$ samples).

We perform:

1. an **insertion** intervention: we retain only the top-$k\%$ most influential pixels of $x^{\text{train}}$, replacing the others with the dataset mean. As a baseline, we retain a random $k\%$ of pixels;

2. a **deletion** intervention: we remove the top-$k\%$ most influential pixels of $x^{\text{train}}$, replacing them with the dataset mean. As a baseline, we remove a random $k\%$ of pixels.

We then perform a single additional SGD step (no momentum, $\text{lr}_{\text{step}}=10^{-3}$) on the loss computed for the masked $x^{\text{train}}$ and measure the change in loss on $x^{\text{test}}$,

$$\Delta\mathcal{L} \;=\; \mathcal{L}_{\text{new}}(x^{\text{test}}) - \mathcal{L}_{\text{old}}(x^{\text{test}}),$$

where more negative $\Delta\mathcal{L}$ indicates a more beneficial update for the test example. Each train–test pair is evaluated under both conditions (Top-$k$ vs. Random), and we report the paired difference $\Delta(T-R)=\Delta\mathcal{L}_{\text{topk}} - \Delta\mathcal{L}_{\text{rand}}$ with normal-approximate $95\%$ confidence intervals (CI) over 1000 pairs per $k$.

If the heatmaps correctly localize influential pixels, then inserting the top-$k$ pixels should yield a strictly more negative test-loss change than inserting random pixels, i.e., $\Delta(T-R) < 0$, and removing the top-$k$ pixels should yield a strictly more positive test-loss, i.e., $\Delta(T-R) > 0$.

**Results** Across a broad range of $k$, Top-$k$ insertion consistently yields more negative test-loss changes than the random control, with statistically significant paired gaps (Table 1). While limited to a single update, this is a good hint at the quantitative effectiveness of the proposed TFA saliency maps obtained by combining grad-cos with Smoothgrad surface important training pixels associated to the prediction on a given test instance. As expected, the effect diminishes as $k$ increases (reduced selectivity) and vanishes at $k=100\%$ by construction.

| $k$ % | insertion | | | deletion | | |
|---|---|---|---|---|---|---|
| | $\mathbb{E}[\Delta\mathcal{L}_{\text{rand}}]$ | $\mathbb{E}[\Delta\mathcal{L}_{\text{topk}}]$ | $\mathbb{E}[\Delta(T-R)] \pm 95\%$ CI | $\mathbb{E}[\Delta\mathcal{L}_{\text{rand}}]$ | $\mathbb{E}[\Delta\mathcal{L}_{\text{topk}}]$ | $\mathbb{E}[\Delta(T-R)] \pm 95\%$ CI |
| 10 | $-0.0829$ | $-0.1560$ | $-0.0730\,[-0.11,\,-0.02]$ | $-0.3475$ | $-0.2805$ | $+0.0670\,[+0.00,\,+0.12]$ |
| 20 | $+0.0831$ | $-0.0782$ | $-0.1613\,[-0.21,\,-0.10]$ | $-0.3532$ | $-0.1677$ | $+0.1854\,[+0.11,\,+0.25]$ |
| 30 | $+0.0739$ | $-0.0422$ | $-0.1161\,[-0.16,\,-0.06]$ | $-0.3301$ | $-0.0882$ | $+0.2418\,[+0.16,\,+0.31]$ |
| 40 | $+0.0504$ | $-0.0523$ | $-0.1027\,[-0.14,\,-0.05]$ | $-0.2978$ | $-0.0544$ | $+0.2434\,[+0.16,\,+0.32]$ |
| 50 | $-0.0086$ | $-0.0661$ | $-0.0576\,[-0.10,\,-0.01]$ | $-0.2455$ | $-0.0629$ | $+0.1826\,[+0.10,\,+0.26]$ |
| 60 | $-0.0312$ | $-0.0857$ | $-0.0545\,[-0.09,\,-0.01]$ | $-0.1882$ | $-0.0548$ | $+0.1334\,[+0.04,\,+0.21]$ |
| 70 | $-0.0786$ | $-0.0995$ | $-0.0208\,[-0.05,\,+0.01]$ | $-0.1330$ | $-0.0935$ | $+0.0395\,[-0.04,\,+0.12]$ |
| 100 | $-0.2175$ | $-0.2175$ | $-0.0000\,[-0.00,\,+0.00]$ | $-0.0995$ | $-0.0995$ | $+0.0000\,[+0.00,\,+0.00]$ |

Table 1: Paired intervention results (CIFAR-10). Negative $\Delta(T-R)$ indicates that Top-$k$ TFA scores insertion improves $x^{\text{test}}$ loss more than a random $k\%$ insertion for the same train–test pair, and positive $\Delta(T-R)$ indicates that Top-$k$ TFA scores deletion degrades $x^{\text{test}}$ loss more than a random $k\%$ deletion.

## 5 USE CASES

### 5.1 EXPLAINING A WRONG PREDICTION

A common diagnostic task in model interpretability is to explain *why* a model makes an incorrect prediction. Our method is well-suited for this, as it identifies not only the training examples most responsible for a given test prediction, but also the specific regions within those training images that contribute to the error.

The practical pipeline is as follows: Given a classification task, suppose a test image is misclassified as class $B$ instead of its correct class $A$. We compute the Grad-Cos scores between the test image and all training images, then sort the training set by these scores. High scores correspond to helpful examples that support correct predictions, whereas low scores reveal the most *harmful* training instances; training on such an image for one additional step would be expected to increase the loss on the test image. In practice, these harmful examples often contain class $B$ in their labels.

To localize the regions in these harmful training images that drive the misclassification, we apply our TFA method. Figure 4 illustrates this process. The test image (a sheep misclassified as a dog) is most harmed by (1) an image of a dalmatian, which visually resembles the sheep, and (2) an image containing both a dog and a sheep, where the influence map shows the model relying on the dog region when predicting the test image.

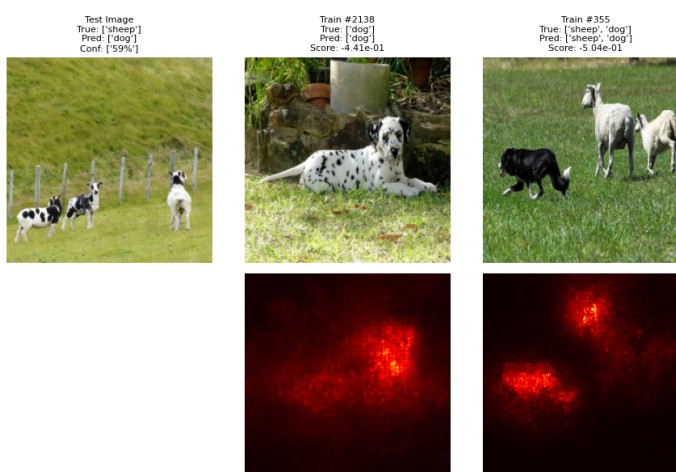

Figure 4: For a test image of a sheep misclassified as dog, the two most influential training images are (1) a dalmatian, and (2) an image containing both a dog and sheep. The influence maps show the model relies on the dog regions when predicting the test image.

### 5.2 DETECTING SPURIOUS CORRELATIONS VIA PATCH-BASED SHORTCUTS

Spurious correlations refer to statistical associations in the training data that do not reflect meaningful or causal relationships, but rather arise due to dataset biases or artifacts. As a result, deep learning

models can base their decisions on such shortcuts, for example by relying on background cues or artificial patterns, instead of learning to recognize the actual object of interest (Izmailov et al., 2022; Xiao et al., 2020). To reveal these hidden biases, it is useful not only to identify which training images most influence a model's predictions, but also to localize the specific regions within those images that drive the decision. Motivated by this, we design a patch-based experiment to assess a model's reliance on a synthetically constructed spurious feature. Specifically, we construct a binary classification task (*sheep* vs. *cow*) and introduce a colored patch (a red square in the bottom right corner) to training images containing sheep, while leaving the validation and test images unaltered. We fine-tune a pretrained ResNet-18 on this biased dataset. The model quickly learns to rely on the presence of the patch as a shortcut for identifying sheep. As a result, during evaluation, the model frequently misclassifies sheep in test images without the patch as cows.

As a comparison to what we would obtain using classical feature attribution, we then apply the off-the-shelf saliency method Grad-CAM (Selvaraju et al., 2017) to a misclassified test image. Grad-CAM primarily highlights the sheep itself, failing to reveal the true cause of the misclassification. This is expected, as the spurious patch is absent from the test image and therefore invisible to methods that only analyze test-time features. Detecting such correlations instead requires examining the training data through training data attribution methods. Using the gradient-cosine similarity, we find that the most influential training images are those containing sheep along with the patch. The corresponding pixelwise saliency maps confirm that the model's predictions are driven largely by the presence of the patch rather than by the animal itself (Figure 5).

Additionally, in Appendix G, a quantitative analysis examining the impact of progressively increasing the proportion of images containing the spurious patch demonstrates that, as the model's reliance on the patch intensifies – resulting in a decrease in classification accuracy for the sheep class – the TFA method correspondingly assigns greater importance to the patch pixels in the training images (Figure 12 in Appendix G).

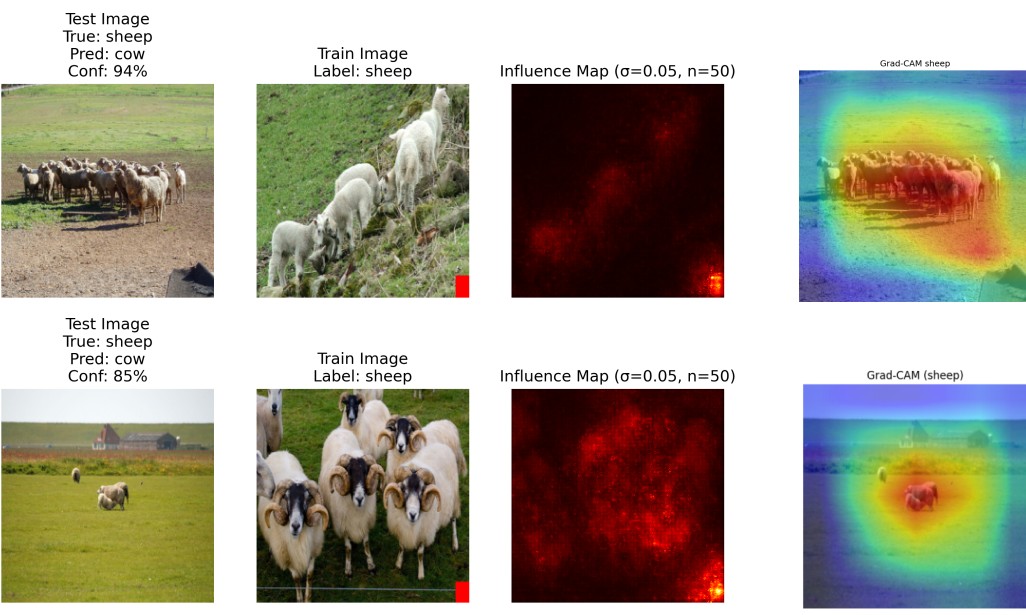

Figure 5: Left to right: (1) Test image of sheep, misclassified as cow; (2) Most influential training image (sheep); (3) Pixelwise influence map reveals the model heavily relies on the red patch for its prediction; (4) Grad-CAM map for the test image.

## 6 CONCLUSION

There exists an intrinsic tension between the growing complexity and opacity of deep learning models (always increasing number of parameters trained on ever-larger datasets), and the rising demand for accountability, reliability, and the ability to provide explanations for model decisions, particularly

in cases of erroneous or biased outcomes. eXplainable AI (XAI) seeks to address this challenge by providing methodologies to interpret trained neural networks, effectively enabling a form of reverse engineering to elucidate their decision-making processes.

In this work, we proposed the training feature attribution (TFA) framework for vision models, designed to trace the patterns utilized during inference back to the specific training examples from which these patterns were learned. We empirically demonstrate that the proposed algorithm, which integrates the grad-cos TDA method with gradient-based FA, generates meaningful saliency maps on training examples. Furthermore, we present two practical use cases illustrating how TFA enhances our understanding of the internal mechanisms underlying model predictions.

Although this work concentrates on pixel-level attributions, the long-term objective is to extend the framework to encompass higher-level, human-interpretable concepts. Such an extension would provide a more abstract and semantically meaningful understanding of the model's learned representations, as well as their origins within the training data.

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

## A   ANALYTIC TOY EXAMPLE FOR TRAINING FEATURE ATTRIBUTION

To illustrate the distinction between training data attribution (TDA) and training feature attribution (TFA), we consider a simple linear ridge regression model in $\mathbb{R}^2$. Because the model admits a closed-form solution, we can compute exact attributions and compare the two decompositions.

**Setup**   We define a linear predictor $f_\theta(x) = \theta^\top x$ with squared loss and $\ell_2$ regularization. The ground-truth rule is $y = x_1 + x_2$, but the training data only reveal this partially: for $i = 1, \ldots, n-1$ we provide samples on the $x_1$ axis,

$$x_i = (x_{i1}, 0), \quad y_i = x_{i1},$$

and a single informative point,

$$x_n = (0, c), \quad y_n = c, \qquad c \neq 0.$$

Let $X \in \mathbb{R}^{n \times 2}$ be the design matrix and $y \in \mathbb{R}^n$ the labels. We train with the ridge objective

$$L(\theta) = \tfrac{1}{2} \sum_{i=1}^{n} (\theta^\top x_i - y_i)^2 + \tfrac{\lambda}{2} \|\theta\|^2, \qquad \lambda > 0.$$

The closed-form solution is

$$\hat{\theta} = (X^\top X + \lambda I)^{-1} X^\top y = \left( \tfrac{S_{11}}{S_{11} + \lambda}, \ \tfrac{c^2}{c^2 + \lambda} \right),$$

where $S_{11} = \sum_{i \neq c} x_{i1}^2$. The Hessian of the objective is diagonal:

$$H = X^\top X + \lambda I = \mathrm{diag}(S_{11} + \lambda, \ c^2 + \lambda).$$

We evaluate predictions at a test point $x_* = (0, t)$.

**TDA (Representer Decomposition)**   In ridge regression, the prediction can be written as

$$f_{\hat{\theta}}(x_*) = \sum_{i=1}^{n} \alpha_i y_i, \qquad \alpha_i = x_*^\top (X^\top X + \lambda I)^{-1} x_i.$$

For $x_* = (0, t)$ and $A = (X^\top X + \lambda I)^{-1} = \mathrm{diag}\left( \tfrac{1}{S_{11} + \lambda}, \tfrac{1}{c^2 + \lambda} \right)$, we obtain

$$\alpha_i = \frac{t}{c^2 + \lambda} \, x_{i2}.$$

Thus $\alpha_i = 0$ for $i \neq n$, and

$$\alpha_n y_n = \frac{tc^2}{c^2 + \lambda} = f_{\hat{\theta}}(x_*).$$

*All predictive mass is attributed to the single informative training example $z_n$.*

**Training Feature Attribution**   We now refine the decomposition down to the level of individual features. For each training example $i$ and feature $k$, we define

$$\beta_{i,k} = x_{ik} \, (e_k^\top A x_*),$$

where $e_k$ is the $k$-th standard basis vector, so that

$$f_{\hat{\theta}}(x_*) = \sum_{i=1}^{n} y_i \sum_{k=1}^{2} \beta_{i,k}.$$

In our toy setup, $e_1^\top A x_* = 0$, hence $\beta_{i,1} = 0$ for all $i$. Meanwhile,

$$e_2^\top A x_* = \frac{t}{c^2 + \lambda}, \qquad \beta_{i,2} = \frac{t}{c^2 + \lambda} \, x_{i2}.$$

Thus $\beta_{i,2} = 0$ for all $i \neq n$, and

$$y_n \beta_{n,2} = \frac{tc^2}{c^2 + \lambda} = f_{\hat{\theta}}(x_*).$$

**Takeaway** Whereas TDA assigns all credit to the single informative training example $z_n$, TFA goes further and reveals that only the second feature of $z_n$ is responsible.

## B    A Sequential TDA→FA Baseline

A natural baseline for TFA is to apply TDA and FA sequentially. One first uses a TDA method to identify influential training examples. Then, for each selected training image, one computes a standard feature-attribution map and uses it as an explanation for its influence on a test point.

A fundamental limitation of this approach is that FA is computed *only* on the training image and therefore does not depend on the test input. The result is effectively a self-attribution map capturing which pixels are important for predicting the training image's own label, rather than which regions influence a specific test example.

To illustrate this issue, we reproduce the setup of Section 4.2, where a single training image contains two semantic classes (e.g., *dog* and *person*). The sequential TDA→FA baseline produces the maps shown in Figure 6.

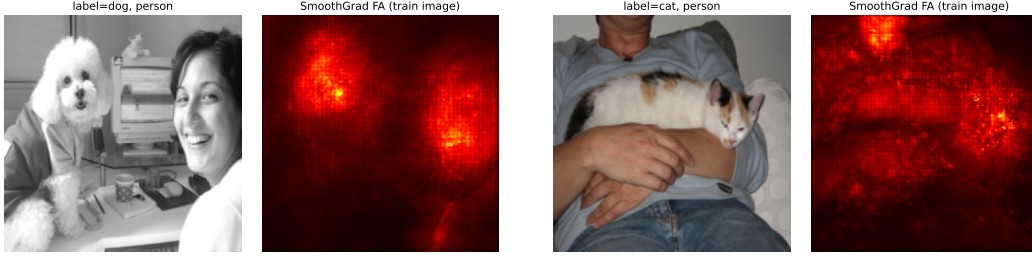

Figure 6: Feature Attribution maps for the training images of Section 4.2. Since FA is applied only to the training image, the sequential TDA→FA baseline is not test-dependent.

Because FA does not incorporate the test image, the resulting attribution maps are identical across test inputs and highlight all salient regions of the training image (here, both classes). Consequently, the sequential TDA→FA baseline cannot capture the test-specific dependencies that TFA provides, making it unsuitable for the use cases considered in this paper.

## C    Ablation study: varying choice of TDA and FA methods

We compare six combinations of training-data attribution (GradCos vs. Influence Functions) and feature-attribution methods (Gradients, SmoothGrad, and Integrated Gradients). The left column shows the test image and the selected training image (selected as the most influential training image based on the gradcos score).

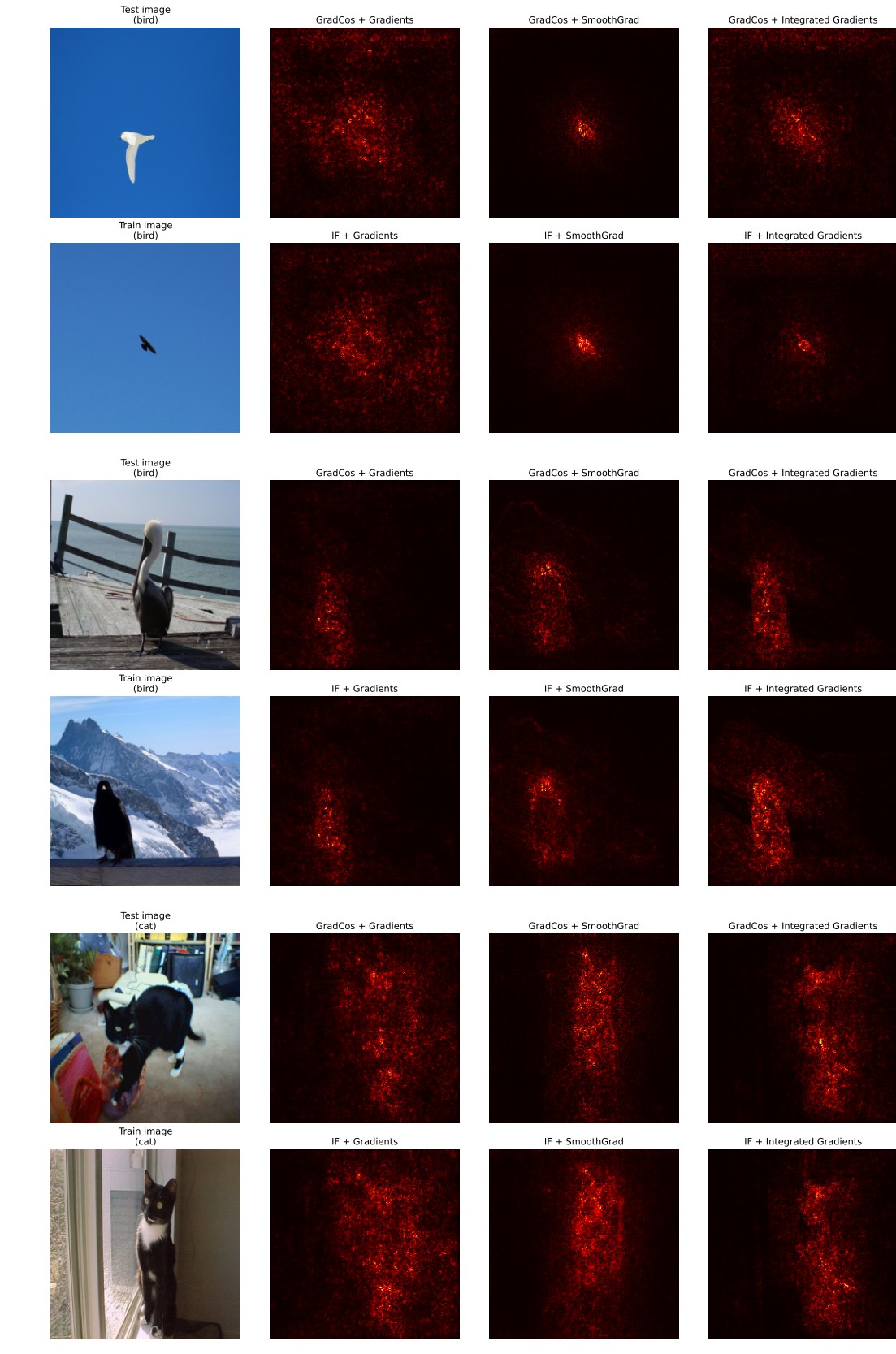

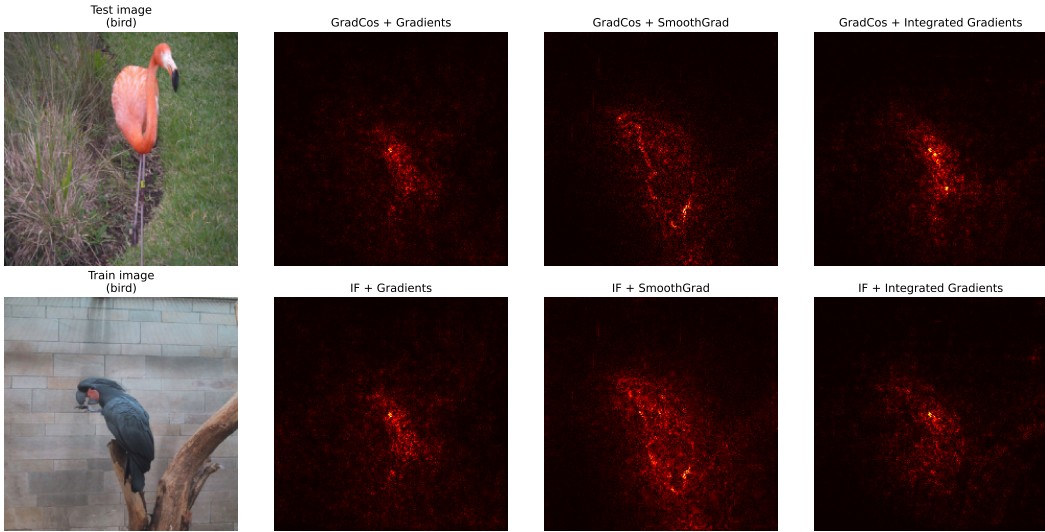

The combination Grad-Cos + SmoothGrad seems to give the best overall results. The problem with integrated gradients is that it heavily depends on the baseline image used. The problem with IF is that it is slow and it sometimes gives outliers (high loss images) as influential (because it doesn't normalize w.r.t the norm of the gradients), see discussion in appendix D below.

## D  FROM INFLUENCE FUNCTIONS TO GRAD-COS VIA RELATIF

As an alternative to grad-cos as choice of TDA method, the influence function (Hampel et al., 1986) from robust statistics quantifies the effect of an infinitesimal upweighting of a training example on a model's output. It was adapted to modern deep learning models in (Koh & Liang, 2017) to estimate the effect of upweighting a single training example $z_i^{\text{train}}$ on the loss of a test example $z_j^{\text{test}}$:

$$I_{\text{IF}}(i, j) = -\nabla_\theta \mathcal{L}(z_j^{\text{test}}; \hat{\theta})^\top H_{\hat{\theta}}^{-1} \nabla_\theta \mathcal{L}(z_i^{\text{train}}; \hat{\theta}),$$

where $H_{\hat{\theta}}$ is the Hessian of the empirical risk at the model parameters $\hat{\theta}$.

In practice, Koh and Liang note that when parameters $\tilde{\theta}$ are obtained via early stopping or in non-convex settings, $H_{\tilde{\theta}}$ may have negative eigenvalues. They address this by replacing $H_{\tilde{\theta}}$ with a damped version $H_{\tilde{\theta}} + \lambda I$, which corresponds to an $L_2$ regularization on the parameters and ensures positive-definiteness.

Our experiments with influence functions were not as satisfactory as expected (Figure 7), which is consistent with a previously reported limitation of influence functions (Barshan et al., 2020; Hanawa et al., 2021), in that the highest-scoring training points for a given test example are often high-loss or atypical samples (e.g., mislabeled data or outliers). Such points tend to appear in the top-$k$ lists for many different test examples, because maximizing $|I_{\text{IF}}(i, j)|$ does not constrain how reweighting $z_i$ affects the model globally. To address this, *Relative Influence Functions* (RelatIF) were proposed, which normalize the influence by the magnitude of the parameter update induced by the training example, thereby emphasizing examples whose effect is more specific to the test point rather than globally dominant (Barshan et al., 2020).

Formally, RelatIF normalizes as follows:

$$S_{\text{RelatIF}, \lambda}(i, j) = -\frac{\nabla_\theta \mathcal{L}(z_j^{\text{test}})^\top (H_{\hat{\theta}} + \lambda I)^{-1} \nabla_\theta \mathcal{L}(z_i^{\text{train}})}{\|(H_{\hat{\theta}} + \lambda I)^{-1} \nabla_\theta \mathcal{L}(z_i^{\text{train}})\|}.$$

In the large-damping regime ($\lambda \gg \|H_{\hat{\theta}}\|$), the inverse can be approximated as:

$$(H_{\hat{\theta}} + \lambda I)^{-1} \approx \frac{1}{\lambda} I,$$

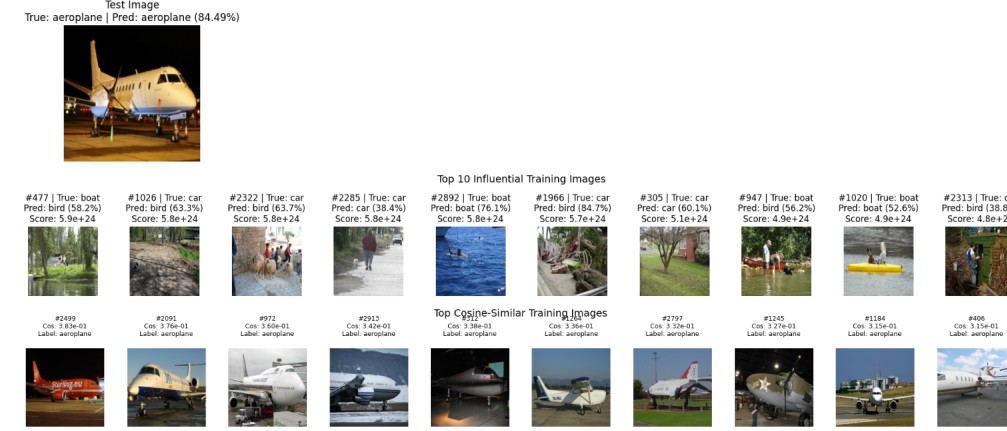

Figure 7: Top-10 most influential training images for a given test image, as identified by influence functions (top) and by Grad-Cos (bottom).

which simplifies the RelatIF score to:

$$S_{\text{RelatIF},\lambda}(i,j) \approx -\frac{\nabla_\theta \mathcal{L}(z_j^{\text{test}})^\top \nabla_\theta \mathcal{L}(z_i^{\text{train}})}{\|\nabla_\theta \mathcal{L}(z_i^{\text{train}})\|} \cdot \frac{1}{\lambda}.$$

This is proportional to the gradient inner product between test and train examples, normalized by the train gradient norm. The *gradient cosine similarity* (Grad-Cos) (Charpiat et al., 2019) is defined as:

$$S_{\text{GC}}(i,j) = \frac{\nabla_\theta \mathcal{L}(z_j^{\text{test}})^\top \nabla_\theta \mathcal{L}(z_i^{\text{train}})}{\|\nabla_\theta \mathcal{L}(z_j^{\text{test}})\| \, \|\nabla_\theta \mathcal{L}(z_i^{\text{train}})\|}.$$

For a fixed test point $j$, the term $\|\nabla_\theta \mathcal{L}(z_j^{\text{test}})\|$ is constant across all $i$, so large-$\lambda$ RelatIF and Grad-Cos produce similar rankings of training examples. Thus, Grad-Cos can be interpreted as the "no-curvature" limit of RelatIF, replacing the Hessian-inverse weighting by a simple directional similarity between gradients. While this interpretation is an approximation, it is reasonable in the large neural networks considered here, where the Hessian is expensive to compute, often ill-conditioned, and in practice dominated by its diagonal structure or noisy low-magnitude eigenvalues. In such settings, removing curvature information tends to yield more coherent attribution scores and explanations.

To illustrate this, Figure 7 compares the top 10 most influential training images for a given test image, as identified by Grad-Cos and by influence functions. While Grad-Cos selects visually similar training samples that align with intuitive, human-understandable explanations, influence functions often return atypical examples that appear to be outliers or high-loss points, offering less interpretable justifications.

## E    DENOISING SALIENCY MAPS WITH SMOOTHGRAD

Pixelwise influence maps, like other gradient-based saliency methods, are often dominated by noise and visually irrelevant fluctuations. While it remains uncertain whether this noise encodes real features of the learned model or simply results from the limitations of the attribution method, its presence can obscure meaningful interpretation. To obtain more robust and interpretable attributions, we combine our method with the SmoothGrad technique Smilkov et al. (2017): we perturb the training image with Gaussian noise, compute the attribution map for each noisy sample, and average the results. The resulting smoothed map is given by

$$\frac{1}{n} \sum_1^n \nabla_{x_i^{\text{train}}} S_{GC}\big(x_i^{\text{train}} + \mathcal{N}(0, \sigma^2 I), z_j^{\text{test}}\big) \tag{3}$$

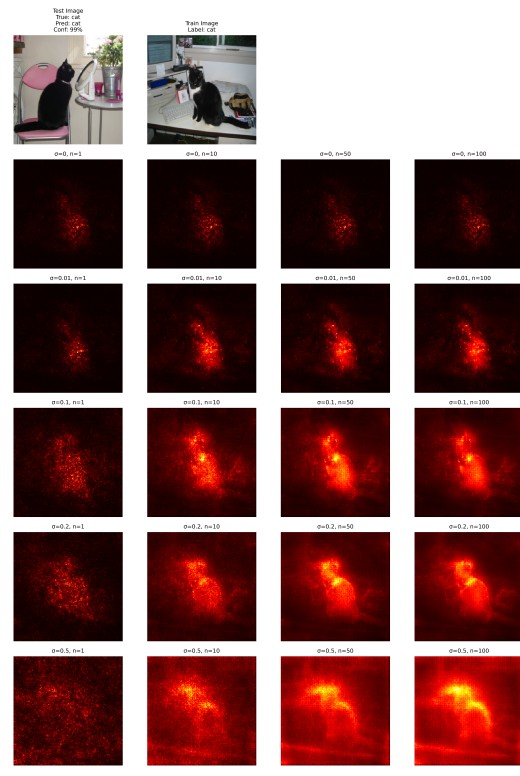

Figure 8: Effect of smoothing parameters on pixelwise influence maps. Each row corresponds to a different noise standard deviation $\sigma \in \{0, 0.01, 0.1, 0.2, 0.5\}$, and each column to a different number of noisy samples $n \in \{1, 10, 50, 100\}$. Shown are the influence maps for a test image (left) predicted as *cat* and its most influential training image (right).

The rationale for this smoothing is that gradients in deep networks, especially those using ReLU activations, can exhibit sharp local fluctuations: small perturbations to the input can cause large, seemingly erratic changes in the gradient, even when the perturbed images appear indistinguishable to a human observer and are classified the same way by the model. These abrupt variations are often not meaningful, but rather artifacts of the model's nonsmooth, piecewise-linear nature Smilkov et al. (2017). By adding Gaussian noise and averaging the resulting saliency maps, we approximate a local average of the gradient field, filtering out these unstable, high-frequency fluctuations while preserving the more robust and informative attributions. For a visualization of the effect of the noise standard deviation $\sigma$ and the number of samples $n$, see Figure 8. Based on these experiments, as well as the recommendations of (Smilkov et al., 2017), we set $\sigma$ to $[5\%, 20\%]$ of the input dynamic range (for images, relative to the pixel intensity scale) and $n \approx 50$, which generally yields robust and interpretable maps.

## F ADDITIONAL EXPERIMENTS

### F.1 ATTRIBUTION TO LAYERS

To analyze how different regions of an image at intermediate network layers influence predictions, we compute gradient-based attributions with respect to the activation map

$$h(x) \in \mathbb{R}^{C \times H \times W}$$

where $C$ is the number of feature channels, and $H \times W$ is the spatial resolution of the activation map at the chosen layer. For a test image $x^{\text{test}}$ and a training image $x_i^{\text{train}}$, we define:

$$\text{Saliency}(x_i^{\text{train}}) = \left| \nabla_{h(x_i^{\text{train}})} \cos\left( \nabla_{h(x^{\text{test}})} \mathcal{L}(z_i^{\text{train}}; \hat{\theta}), \ \nabla_{h(x_i^{\text{train}})} \mathcal{L}(z_i^{\text{train}}; \hat{\theta}) \right) \right|$$

where:

- $\nabla_{h(x)}$ denotes the gradient with respect to the activation map $h(x)$ at the chosen layer.
- The cosine similarity measures the alignment between the influence of test and training samples through that layer.

Averaging over channels yields a 2D spatial saliency map:

$$\text{Saliency2D} = \frac{1}{C} \sum_{c=1}^{C} \text{Saliency}_{c,:,:}$$

For example, for `layer3` in ResNet-18, $(H, W) = (14, 14)$. The resulting map is then upsampled (e.g., bilinear interpolation) to match the input resolution (e.g., $224 \times 224$).

Figure 9 illustrates the outputs of this approach. As we move to deeper layers, the highlighted regions of the saliency maps appear smoother, which is expected since the maps are obtained by upsampling from progressively smaller activation maps. Nonetheless, the highlighted object remains consistent across layers, even when compared to the raw saliency map computed with respect to the training image. For the last convolutional layer (`layer4`), however, the focus on the object decreases and the most salient pixels extend over a larger portion of the image. This observation is consistent with the results reported in the Grad-CAM paper (Selvaraju et al., 2017) when experimenting with ResNets.

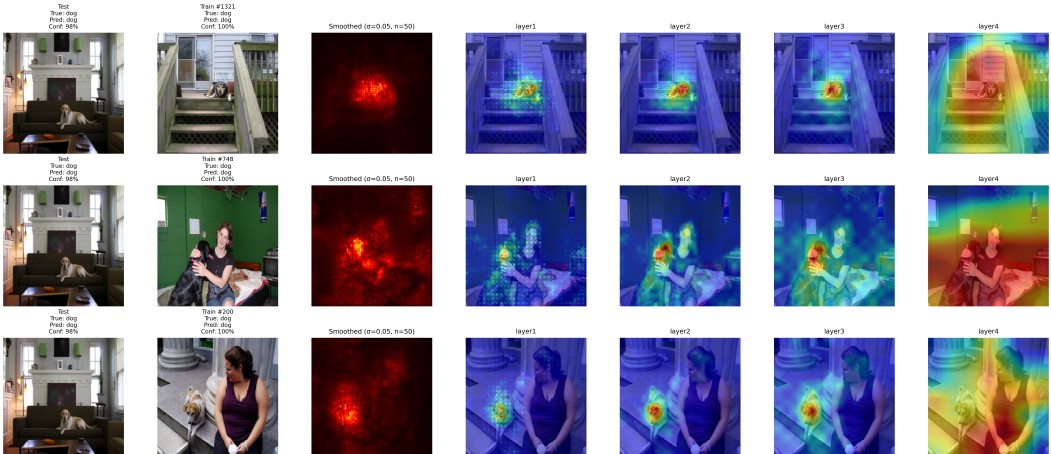

Figure 9: From left to right: test image, training image that includes a dog, pixelwise smoothed Grad-Cos saliency map ($\sigma$=0.05, $n$=50), and layerwise Grad-Cos saliency maps. Each layerwise map corresponds to the last convolutional layer in each residual block of the ResNet-18 architecture.

### F.2 VARYING MODELS

To assess how architectural differences affect our pixelwise influence maps, we compare three backbones under the same training and preprocessing protocol: a ResNet-18 and a ResNet-50 (He et al., 2016), and a ViT-B/16 (Dosovitskiy et al., 2020) (all pretrained on ImageNet and fine-tuned on Pascal VOC 2012). For each test-train pair, we compute Smoothed Grad-Cos maps (Eq. 3) with $\sigma$=0.1 and $n$=20 noisy samples.

We observe (Figure 10) that the Vision Transformer consistently produces heatmaps with a visible patchwise structure. This effect stems from its architecture: ViT processes images as a sequence of non-overlapping patches (here $16 \times 16$ pixels), which are flattened and linearly projected into patch tokens. When computing gradients with respect to the input image, the backpropagation signal flows through this patch embedding step, so gradients are computed independently within each patch. As a result, the effective spatial resolution of the influence map is limited by the patch size, and channel-aggregated attributions often appear uniform within each patch.

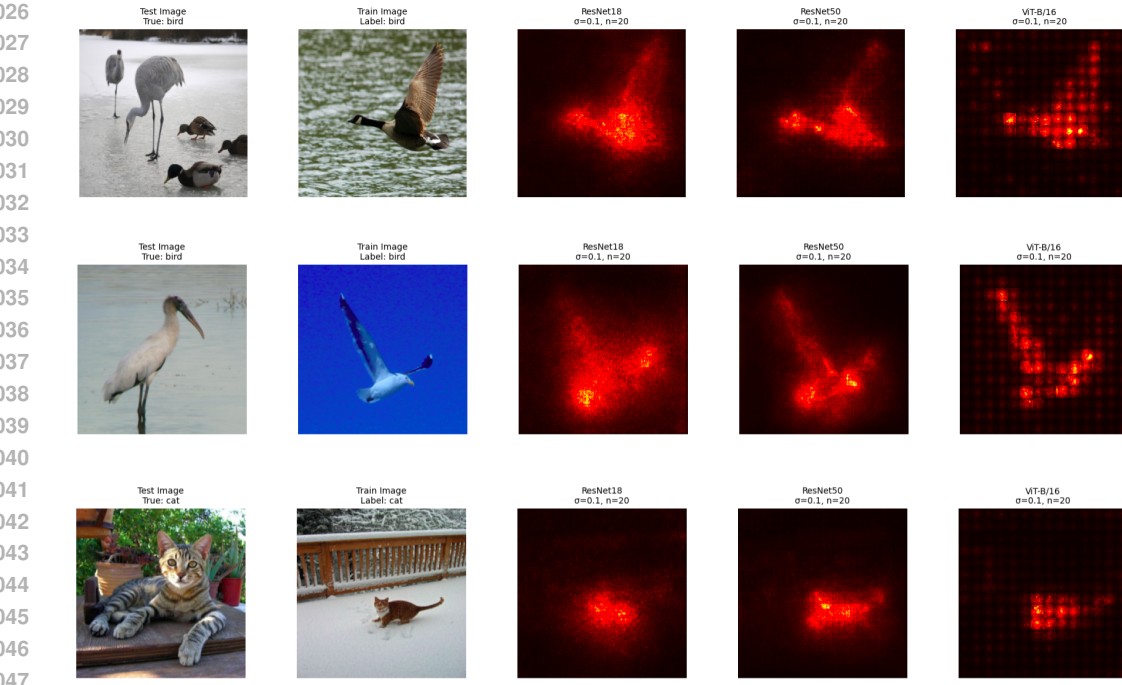

Figure 10: Influence comparison across backbones. Each row shows (from left to right): test image, a training image from the same class, and influence maps produced by ResNet-18, ResNet-50, and ViT-B/16 ($\sigma$=0.1, $n$=20).

Interestingly, we find that ResNet-50 is slightly less sensitive to noise compared to ResNet-18, likely due to its deeper architecture and larger receptive fields.

## G  QUANTITATIVE EVALUATION OF SHORTCUT DETECTION ON CIFAR-10

To quantitatively assess how effectively our method detects spurious correlations, and how strongly the model relies on them as a function of their prevalence in the training set, we design a controlled patch-based shortcut experiment.

We construct a subset of CIFAR-10 (Krizhevsky, 2009) containing three classes: *dog*, *ship*, and *automobile*. A square colored patch is inserted in the lower-right corner of a fixed proportion of the training images labeled as *dog*. This proportion (referred to as the *patch fraction*) denotes the percentage of *dog* training images that are patched, and we vary it across 17 values from 0% to 100%. For each patch fraction, we train the same lightweight CNN from scratch on the corresponding training set.

At evaluation time, to probe shortcut reliance, we also create a "patched-ship" test set by inserting the same patch into ship test images; all other test images remain unmodified. We then report: overall test accuracy, accuracy on unpatched *dog* images, and accuracy on patched *ship* images (to test whether the model associates the patch with the *dog* class).

As qualitative examples at four patch fractions (0%, 5%, 85%, 95%), see Figure 11.

To verify whether our method localizes the shortcut, we apply training feature attribution to the patched-ship images. If the shortcut has been learned, these probes are increasingly misclassified as *dog*, and the influence maps should highlight the patch region. For each patch fraction, we sample five patched-ship test images, identify the ten most harmful training images (most oppositely aligned gradients), and compute pixelwise influence maps. We keep the top 10% most salient pixels and measure the proportion falling inside the patch (patch attribution fraction).

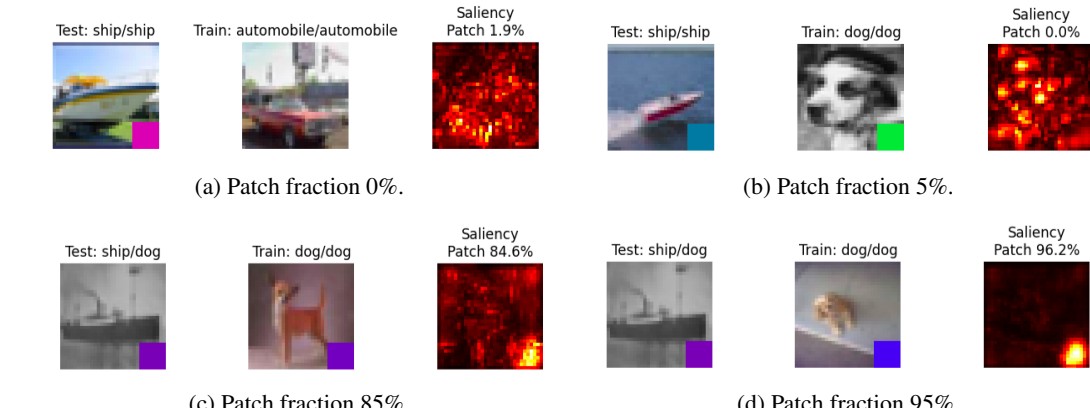

(a) Patch fraction 0%.          (b) Patch fraction 5%.

(c) Patch fraction 85%.          (d) Patch fraction 95%.

Figure 11: Qualitative triplets for the CIFAR-10 shortcut experiment at four patch prevalences. Each panel shows (left to right): the test image with true/predicted label, the most harmful training image, and the pixelwise influence map with the percentage of top-10% saliency inside the patch.

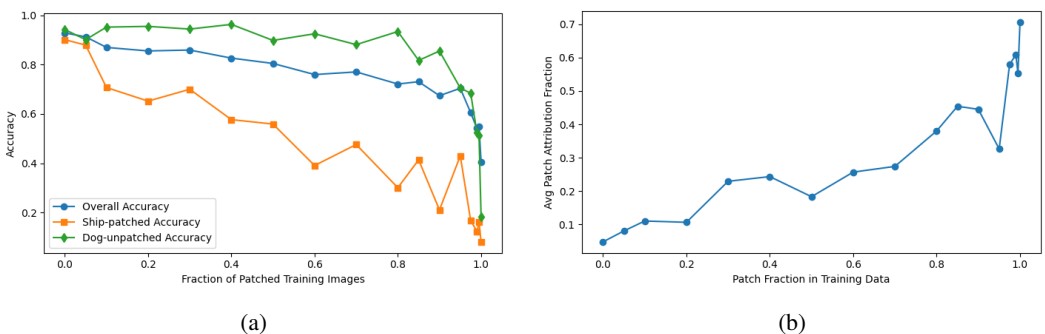

(a)          (b)

Figure 12: (a) Performance as the patch prevalence increases; (b) Localization of the shortcut via training feature attribution .

As shown in Figure 12a, as the fraction of *dog* training images with the patch increases, the accuracy on *patched ship* and *unpatched dog* images decreases, indicating that the model has adopted the patch as a shortcut. This effect is more pronounced for patched ships, which never co-occur with the patch during training and are thus quickly misclassified as dogs, whereas unpatched dogs remain recognizable until the patch prevalence becomes extreme. Consistently, Figure 12b shows a rising patch-attribution fraction, confirming that training feature attribution increasingly localizes to the patch region as its proportion grows.

