# OpenReview forum: "Training Feature Attribution for Vision Models"
_ICLR.cc/2026/Conference — Submitted to ICLR 2026_

### Official Review · Reviewer_AcPD · 2025-10-30

**Soundness:** 2
**Presentation:** 3
**Contribution:** 2
**Rating:** 2
**Confidence:** 4

**Summary:**

The paper introduces *Training Feature Attribution (TFA)*, a method that integrates *training data attribution (TDA)* with *feature attribution (FA)*. TDA identifies which training samples most influenced a given test prediction, while FA highlights which input features of the test sample were most relevant for that prediction. By combining these approaches, TFA identifies influential training examples *together with* the specific input regions that contributed to the model’s decision.

To accomplish this, the method uses *gradient cosine similarity* (Charpiat et al., 2019) to score training samples by their relevance to a test prediction. Since the full pipeline remains differentiable, a gradient-based feature attribution method is then applied on this measure, enabling attribution over training data.

The paper demonstrates the interpretability benefits of TFA through qualitative examples and provides quantitative results showing that the method significantly outperforms random attribution baselines.

**Strengths:**

- The paper clearly identifies a gap in related work and motivates why combining TDA and FA is useful.
- The writing is clear, structured, and easy to follow.
- The mathematical foundations are well connected to intuitive explanations.
- The paper offers an empirical evaluation, at least to some extent.

**Weaknesses:**

- **W1 A simple baseline is missing.** A trivial approach to the problem would be to use TDA to select the most relevant training samples, then pass those samples through the model to compute standard FA. While the paper’s joint TFA approach is theoretically motivated, it is unclear whether similar empirical results could be achieved with this simpler baseline. Including such a comparison would help strengthen the empirical case for TFA.
- **W2 Similar insights might be achievable with existing or trivial explanation methods.** For example, the observations in Sec. 5.2 may be attainable using counterfactual explanations, and the findings in Sec. 5.1 might also emerge from the simple baseline described in W1. If these methods provide similar insights, it raises the question of whether TFA is necessary. The paper should empirically demonstrate that TFA offers insights that go beyond those provided by existing or trivial baselines.

- **W3 The evaluation is relatively limited.** Given that the proposed method conceptually combines two existing techniques, a more comprehensive evaluation would be expected to show the added value of this integration. While the theoretical justification is clear, the empirical results do not fully reflect this strength. Moreover, much of the evaluation relies on a small number of qualitative examples, making it difficult to assess how broadly the method generalizes.

**Questions:**

- How does the proposed method compare to the trivial baseline proposed in W1? An explicit comparison would clarify whether the joint formulation provides meaningful advantages over the straightforward two-step approach.
- Are there scenarios or case studies where TFA clearly yields insights beyond existing explanation methods? If so, it would be valuable to empirically demonstrate this using a diverse set of baselines, covering different explanation families such as FA, TDA, counterfactual explanations, and others. This would help establish when and why TFA should be preferred in practice.

Lastly, I thank the authors for their effort and look forward to the rebuttal. If my concerns are sufficiently addressed and none of the other reviewers raises critical issues, I would be happy to reconsider and increase my score.

---

> ### Author Response · Authors · 2025-11-24
>
> We are grateful for your comprehensive review and constructive suggestions. In addition to our general response and the revised PDF, please find below our answers to the specific points you raised:
>
> **W1: Simple TDA + FA applied to predicting the training class:**
> We have added this baseline in Appendix B. A key limitation is that the saliency map does not depend on the test image; it highlights all elements used for predicting the test class but does not provide insights into the specific features learned from the training image. This issue is particularly evident in the case of multiple classes per image: all classes of the training image will be highlighted, even if the test image only contains one of the classes. Please refer to the discussion in Appendix B of the paper.
>
> **W2: Use cases could be solved using either TDA or FA:**
> Our main point is that combining TDA and FA is more general than using either approach alone, as it provides insights across all use cases simultaneously. This avoids the need to resort to specific methods tailored to each use case. In this sense, we believe TFA should be a default choice, as it subsumes both TDA and FA.
>
> **W3: Evaluation is limited:**
> We have strengthened our evaluation by adding a more comprehensive ablation study in Appendix C. While the concept of TFA is general and compatible with other TDA/FA pairs, our work suggest using Grad-Cos and SmoothGrad due to their computational efficiency and their passing of the sanity checks proposed in [1] and [2].
>
> [1] Julius Adebayo, Justin Gilmer, Michael Muelly, Ian Goodfellow, Moritz Hardt, and Been Kim. Sanity checks for saliency maps. Advances in neural information processing systems, 2018
>
> [2] Kazuaki Hanawa, Sho Yokoi, Satoshi Hara, and Kentaro Inui. Evaluation of similarity-based explanations. In International Conference on Learning Representations, 2021

---

> > ### Comment · Reviewer_AcPD · 2025-11-26
> >
> > I thank the authors for their reply.
> >
> > **Re: W1** Thank you for providing Appendix B. I am sorry if my original recommendation was not sufficiently clear. I did not mean applying FA to all classes present in the retrieved training samples, but specifically to the class of the *test* sample, which is straightforward to implement. I would like to see a comparison of *this* trivial baseline to the proposed method. If possible, it would also be helpful to evaluate this comparison quantitatively and not only on a small set of qualitative examples.
> >
> > **Re: W2** I am not convinced by the claim that TFA should be a default choice simply because it subsumes both TDA and FA. In the provided examples, attribution for the test sample itself is never computed. This becomes problematic when the test image is slightly out-of-distribution and contains important features not represented in the training data. Here, standard FA would still be necessary to analyze the test sample.
> >
> > That said, I do understand the theoretical motivation that combining TDA and FA can offer a more complete picture. However, as also noted by Reviewer waGT, the proposed method is mostly a combination of established TDA and FA components. While this is generally fine, I would expect the paper to demonstrate a clear advantage in scenarios where precisely this *combination* is needed, rather than in a sequence of examples where either TDA, FA, or other existing methods would suffice.
> >
> > Somewhat exaggeratedly, one could say that any two XAI methods could be combined and declared the new default since their combination inherits the advantages of both. However, if we do not have an application where the combination is truly necessary—and not just each method individually—it is difficult to see the added value for the community.
> >
> > **Re: W3** Thank you for adding the additional ablation studies. Given the points above, what I was primarily looking for is a quantitative evaluation showing that the proposed method offers advantages over the trivial baseline from W1, and an experiment demonstrating that *combining* TDA and FA is *necessary* for certain tasks.

---

> > > ### Author Response · Authors · 2025-12-03
> > >
> > > Thank you again for the follow-up and for clarifying your expectations. Below we address your remaining concerns.
> > >
> > > We implemented the baseline you proposed (TDA then FA on the training image using the test-class logit) and compared it to TFA using the same insertion protocol as in our quantitative experiment.
> > >
> > > Below are the paired statistics (mean test-loss change for each kept-pixel ratio \(k\)). (R): random pixels, (T): TFA, (FA): the baseline. (more negative is better)
> > >
> > > | k (%) | E[R]    | E[T]    | E[FA]   | E[T−FA]  |
> > > |-------|---------|---------|---------|----------|
> > > | 10    | -0.2774 | -0.3289 | -0.2919 | -0.0370  |
> > > | 20    | -0.3413 | -0.3062 | -0.2879 | -0.0184  |
> > > | 30    | -0.3457 | -0.4118 | -0.3570 | -0.0548  |
> > > | 40    | -0.3455 | -0.4225 | -0.3894 | -0.0331  |
> > > | 50    | -0.3647 | -0.3955 | -0.3686 | -0.0269  |
> > > | 60    | -0.3718 | -0.4106 | -0.3815 | -0.0290  |
> > > | 70    | -0.3832 | -0.3980 | -0.3897 | -0.0083  |
> > > | 100   | -0.5071 | -0.5071 | -0.5071 | -0.0000  |
> > >
> > > We observe that:
> > >
> > > - Both TFA and the FA-on-train baseline are generally more effective than random pixel selection, confirming that the retrieved training regions carry meaningful signal.
> > > - TFA tends to produce stronger insertion effects (more negative changes) than the baseline, although the effect size remains modest.
> > >
> > > A key limitation of the FA-on-train baseline is that its saliency map on the training image does not depend on the specific test instance beyond its class label: it highlights all regions that support the test class, but does not distinguish which of these regions actually drive the prediction for a given test image. One scenario where this matters is when a training image contains several instances of the same class, while the test image shows only one of them. The FA-on-train baseline, driven by the class logit, tends to highlight all class-relevant instances in the training image. In contrast, TFA is driven by a similarity score computed from the test loss gradient, and can therefore emphasize the training region whose features are most similar to those of the test instance.

---

### Official Review · Reviewer_waGT · 2025-11-01

**Soundness:** 3
**Presentation:** 2
**Contribution:** 2
**Rating:** 2
**Confidence:** 4

**Summary:**

Summary
The paper introduces training feature attribution (TFA) that attributes a test prediction to spatial regions of specific training images by differentiating a gradient-cosine (grad-cos) training–test similarity score with respect to the training image, with qualitative demonstrations on Pascal VOC, a CIFAR-10 insertion study, and use cases on error analysis and spurious correlations.
The approach is positioned as unifying training-data attribution with feature attribution to answer “which parts of which training images most influenced this test prediction” in vision models.

Soundness
The core estimator—taking input gradients of a differentiable TDA scalar (grad-cos) w.r.t. a training image—is methodologically sound and consistent with standard saliency practices, and the analytic ridge-regression example supports the intuition at a toy scale.
However, the causal validation relies on single-step SGD updates on masked training images rather than full or multi-epoch retraining, which weakens claims about training-time causal importance. Quantitative faithfulness is limited to insertion tests without complementary deletion or broader sanity checks, leaving robustness underexplored.

Presentation
The paper is clearly written, well structured, and easy to follow, with intuitive framing and informative qualitative figures that aid understanding of TFA’s outputs. That said, plots and comparisons feel thin: key baselines (e.g., TracIn/representer as TDA backbones) are absent, and quantitative panels do not sufficiently probe stability, sensitivity, or scale, making the empirical evidence less convincing than the narrative suggests.

Contribution
The contribution is incremental: it instantiates a straightforward combination of existing training-data attribution (via grad-cos) and feature attribution (via input gradients with optional SmoothGrad) rather than proposing a novel attribution principle, estimator, or theory with guarantees. Practical value is moderate for error analysis and uncovering spurious correlations, but the lack of stronger evaluations and scalability studies limits impact at scale.

**Strengths:**

-Clear problem statement linking example-level and feature-level explanations, producing test-specific training-region maps that are easy to interpret qualitatively.
-Simple and implementable pipeline (grad-cos + input gradients + SmoothGrad) with demonstrations on common datasets and architectures, plus an analytic toy example for intuition.
-An insertion study on CIFAR-10 shows TFA-selected training pixels outperform random selections across multiple k, offering initial quantitative support.

**Weaknesses:**

-Limited novelty: primarily a composition of known TDA and FA components without a new estimator, objective, or theoretical advance.
-“Masking and retraining” claim is operationalized as a single-step update, not full retraining, weakening causal interpretation and overstating empirical claims.
-Missing comparisons to alternative TDA backbones (e.g., TracIn, representer point methods) and absent deletion tests or comprehensive sanity checks for attribution faithfulness.
-No runtime/memory profiling or scaling analysis for large datasets, leaving practical feasibility and batching strategies unclear.

**Questions:**

-Do the authors perform any full or multi-epoch retraining after masking salient training regions, or is the evaluation limited to single-step updates; if only the latter, can claims and terminology be aligned accordingly ?

-How does TFA scale to ImageNet-sized corpora in terms of wall-clock time and GPU memory when considering many train–test pairs, and what batching or approximate retrieval strategies are used ?

-How do results change if the TDA backbone is swapped for TracIn or representer methods, both qualitatively and via insertion/deletion metrics and stability analyses ?
-Can the authors include deletion ablations, gradient-sanity checks, and sensitivity to SmoothGrad hyperparameters to strengthen faithfulness claims ?

---

> ### Author Response · Authors · 2025-11-24
>
> Thank you for your detailed feedback and insightful recommendations. In addition to our general response and the revised PDF, please find below our answers to the specific points you raised:
>
> **Quantitative evaluation / Single-step update:**
> We have added the deletion experiment you suggested in Section 4.3. We acknowledge that evaluating TDA methods remains an open challenge; for example, approaches like [1] require training numerous models from scratch. Our evaluation is significantly more lightweight, as it inherits the inherent difficulty of accurately evaluating both FA and TDA, but without the need for extensive retraining, we have thus moderated the claims accordingly in 4.3.
>
> **Scaling / Compute complexity of TFA:**
> The most computationally intensive step for scaling TFA to datasets like ImageNet involves applying TDA to all training examples to identify high-influence instances. This challenge is not specific to TFA. In comparison, the FA component is very inexpensive, requiring only a few gradient computations. Our experiments on Pascal VOC, which has fewer examples but larger images than datasets like CIFAR-10/100, demonstrate that using Grad-Cos as the TDA component is much cheaper, requiring only a single gradient computation per training image. Both influence functions and Grad-Cos are viable options, though the debate on which TDA method to prefer remains open. We acknowledge the efforts in [2] to scale influence functions to LLM pretraining corpora.
>
> **Representer and TracIn:**
> Representer scores could, in theory, be used for TFA, although we did not find it straightforward to extend them to multiclass settings. TracIn, on the other hand, requires storing checkpoints throughout training, rendering the approach less practical for debugging trained networks. Still, in theory, this should also be usable as the TDA component.
>
> **Deletion ablation:**
> These results are now included in Section 4.3.
>
> **Gradient-sanity checks:**
> Our approach inherits the properties of the chosen FA method. We selected SmoothGrad because it performs competitively in [3] in terms of gradient-sanity checks.
>
> **Sensitivity to SmoothGrad hyperparameters:**
> A sensitivity analysis is provided in Appendix E. The results align with the original SmoothGrad paper. We recommend setting the noise level ($\sigma$) to between 5% and 20% of the input dynamic range (relative to pixel intensity for images) and using 50 noisy samples, which generally yields robust and interpretable saliency maps.
>
> [1] Park, S. M., Georgiev, K., Ilyas, A., Leclerc, G., & Madry, A. TRAK: Attributing Model Behavior at Scale. In International Conference on Machine Learning, 2023
>
> [2] Roger Grosse, Juhan Bae, Cem Anil, Nelson Elhage, Alex Tamkin, Amirhossein Tajdini, Benoit
> Steiner, Dustin Li, Esin Durmus, Ethan Perez, et al. Studying large language model generalization with influence functions. arXiv:2308.03296, 2023
>
> [3] Julius Adebayo, Justin Gilmer, Michael Muelly, Ian Goodfellow, Moritz Hardt, and Been Kim. Sanity checks for saliency maps. Advances in neural information processing systems, 2018

---

> > ### Comment · Area_Chair_A4fK · 2025-11-28
> >
> > Dear Reviewer,
> >
> > Please make sure you read the authors' response and engage with them in the discussion before the end of the discussion period on **Dec 03 '25 09:00 PM UTC**. This is a hard deadline.
> >
> > Thank you for supporting quality peer review at ICLR.
> >
> > AC

---

### Official Review · Reviewer_Mhg7 · 2025-11-01

**Soundness:** 3
**Presentation:** 3
**Contribution:** 2
**Rating:** 4
**Confidence:** 4

**Summary:**

This paper introduces a training feature attribution that links test predictions to regions of specific training images.
Specifically, they combine the gradient cosine similarity TDF with a gradient-based feature attribution methods.

The paper includes a quantitative analysis of their method’s performance by comparing the loss changes after one additional training step: For this, they either mask everything apart from (i) the k most influential input pixels or (ii) k random pixels on the most influential training samples corresponding to a set of test images.
 Further, they provide evidence that their method can identify harmful examples driving misclassifications and reveal spurious correlations.

**Strengths:**

Overall the paper is clearly written. Further, the combination of training data attribution with feature attribution seems like a useful approach to analyse image models. The two use cases show interesting possible applications of the method.

**Weaknesses:**

Combining a relevant set of training images, e.g. maximally activating training samples for a certain neuron, with feature attribution methods like GradCAM is not uncommon in the explainability literature. I suggest the authors discuss the connections of their approach to such methods. E.g. [1] and [2] utilise frameworks based on max. activating training images and GradCAM saliency maps to identify spurious correlations in the training data.

Related to the previous point, the paper could provide more ablations on their choices of (i) the TDA method and (ii) the feature attribution method. While it is a positive result, that the introduces method outperforms random saliency, this experiment could be extended to compare to simple baselines, e.g. most similar images based on LPIPS or CLIP similarities as TDA, as well as other kinds of saliency maps.

[1] Salient ImageNet, Singla and Feizi, 2022, https://arxiv.org/pdf/2110.04301

[2] Spurious Features Everywhere, Neuhaus et al, 2023, https://arxiv.org/pdf/2212.04871

**Questions:**

1. How does the choice of the smoothing hyperparameters (number of noise samples and standard deviation) affect your quantitative results?

2. How many of the training images included the squares in the results shown in Figure 5?

3. In Figure 5, how would the GradCAM compare when computed on the train image instead of the test image, i.e. would it also highlight the square?

---

> ### Author Response · Authors · 2025-11-24
>
> We appreciate your thorough review and valuable suggestions. In addition to our general response and the revised PDF, please find below our answers to the specific points you raised:
>
> **Choice of smoothing hyperparameter of SmoothGrad**
> A sensitivity analysis is provided in Appendix E. The results align with the original SmoothGrad paper. We recommend setting the noise level ($\sigma$) to between 5% and 20% of the input dynamic range (relative to pixel intensity for images) and using 50 noisy samples, which generally yields robust and interpretable saliency maps.
>
> **Number of training images in the square spurious correlation experiment:**
> A sensitivity analysis is included in Appendix G, which demonstrates that reliance on the spurious feature (the squares) increases as the proportion of training examples containing it grows (see Figure 12). We also would like to point out that we used relatively large squares in this experiment to ensure visibility for the reader. However, we have conducted additional experiments with smaller patches, which are much more challenging to detect without the aid of TFA.
>
> **Would GradCAM applied to a training image containing the patch also highlight it?**
> Yes, it is very likely, since the model has learned to associate the patch with the class. Our main point is that combining TDA and FA is more general than using either approach alone, as it provides insights across all use cases simultaneously. This avoids the need to resort to specific methods tailored to each use case. In this sense, we believe TFA should be a default choice, as it subsumes both TDA and FA.
>
> **Provide more ablations in the choices of FA and TFA:**
> We have strengthened our evaluation by adding a more comprehensive ablation study in Appendix C. While the concept of TFA is general and compatible with other TDA/FA pairs, our work suggest using Grad-Cos and SmoothGrad due to their computational efficiency and their passing of the sanity checks proposed in [1] and [2].
>
> [1] Julius Adebayo, Justin Gilmer, Michael Muelly, Ian Goodfellow, Moritz Hardt, and Been Kim. Sanity checks for saliency maps. Advances in neural information processing systems, 2018
>
> [2] Kazuaki Hanawa, Sho Yokoi, Satoshi Hara, and Kentaro Inui. Evaluation of similarity-based explanations. In International Conference on Learning Representations, 2021

---

### Official Review · Reviewer_E8eB · 2025-11-02

**Soundness:** 3
**Presentation:** 4
**Contribution:** 3
**Rating:** 6
**Confidence:** 3

**Summary:**

The paper builds on and combines two different attribution paradigms from previous work: feature attribution (FA), which explains individual predictions by highlighting the most important pixels in a given test image, and training data attribution (TDA), which identifies the training examples that most strongly influence a model’s prediction for that test image. This paper aims to unify these two components: given a test image, the proposed method not only identifies the most influential training samples (as in TDA) but also determines which regions within those training images contributed most to the prediction.

To identify relevant training samples (as in TDA), the approach follows the grad-cos attribution formulation. To additionally localize the salient regions, the method differentiates the grad-cos score w.r.t. the training image itself, which allows tracing down the impact to individual pixels, in a similar manner as a standard feature attribution.

The evaluation is primarily performed in a qualitative manner, demonstrating how given a test image, the attribution focuses on the correct object samples and regions within the training data. An additional quantitative insertion intervention shows that the reported saliency map indeed highlights regions with an above-average impact on the prediction. Finally, the paper provides two practical use cases, which demonstrate in partially synthetically crafted proof-of-concept experiments that the proposed method could be used to explain mispredictions or to detect spurious correlations.

**Strengths:**

Taking the gradient of the grad-cos attribution formulation with respect to the training samples — to the best of my knowledge —  represents a new direction in explainable AI, as it extends attributions beyond the test domain to the underlying training material itself. Rather than identifying which pixels in a test image impact a prediction, it allows uncovering which regions in specific training examples are most responsible for the prediction. The paper motivates the relevance of this extension by clearly presented, understandable qualitative attribution maps and two practical use cases.

The insertion-based intervention experiment serves as a sanity check, confirming that the highlighted regions indeed contain more predictive signal than random patches. Notably, the smoothing hyperparameter choice is examined in the supplementary material by an additional ablation study.

The paper is very well written and is easy to follow.

**Weaknesses:**

In contrast to the well-curated qualitative assessment of the proposed approach, the quantitative evaluation remains very limited: the provided insertion-based test captures only a single aspect of the attribution’s faithfulness and does not fully support the qualitative findings, e.g., the correct attribution object focus in Figure 3. In particular, given that segmentation masks are available for the Pascal VOC dataset used in the experiments, the qualitative results could have been easily complemented with quantitative localization metrics. Measuring the overlap between the attributed regions and the corresponding target masks could’ve ruled out concerns about potential sample selection biases.

As pointed out by the paper, there is some overlap in terms of goals between the proposed approach and prototypes, which also highlight relevant parts of training images. While there are differences, e.g. in that TDA can be applied to pretty much any trained neural network, I was missing a more in-depth discussion of the similarities and differences. Also it would be interesting to see if TFA applied to prototype networks lead to similar parts in the training data being identified.

Minor points:
* The mathematical notation in Sec. 3.1 and Sec. 2 is not completely consistent. This does not distract too much, but it would be nicer if it was unified.
* When the paper talks about data (e.g., l. 098), it would be good to always clarify whether training or test data is meant, since this is an important distinction here.
* The title can be read in two different ways (after having read the paper, it is of course clear, but not necessarily before). It could be misread as “Training (Feature Attribution)”, i.e. an approach that trains feature attribution methods. This could be avoided by something like “Attribution of training features…” or something like this, but then the abbreviations do not work out as nicely. Nothing critical, just something for the authors to reflect upon.

**Questions:**

Given the limited quantitative evaluation, I am a little bit on the fence regarding this paper. Strengthening the paper through more quantitative results would likely make me consider raising my score.

* Please provide additional quantitative evaluations to complement the strong qualitative results. Since the VOC dataset includes segmentation masks, would it be possible to state localization-oriented metrics (e.g., IoU, F1) to support the qualitative findings in Figure 3?
* Beyond the insertion test, consider adding quantitative measures that evaluate the spatial correspondence between attributed regions and target objects, as well as comparisons against random or background baselines.
* Could you discuss the relation to prototype approaches in more detail?

---

> ### Author Response · Authors · 2025-11-24
>
> We would like to thank you for your detailed review and suggestions, as well as your overall positive review. In addition to our general response and the revised PDF, please find below our answers to the specific points you raised:
>
> **Limited evaluation:**
> We have strengthened our evaluation by adding a more comprehensive ablation study in Appendix C. While the concept of TFA is general and compatible with various TDA/FA pairs, we recommend using Grad-Cos and SmoothGrad due to their computational efficiency and their passing of the sanity checks proposed in [1] and [2]. Additionally, we have included a deletion test to complement the insertion test in the quantitative evaluation presented in Section 4.3.
>
> **Prototypes:**
> Since TFA identifies similarities between parts of pairs of images, our next goal is to leverage TFA as a backbone method to automatically discover prototypical parts within training images. This is similar to approaches such as those in [3], where patterns used at test time are learned from training images. However, the precise methodology for doing so is not yet fully established.
>
> **Evaluation with respect to segmentation masks:**
> We agree that evaluating TFA using segmentation masks could be another valuable approach. Due to time constraints and to the fact that it is not clear what exactly to compare, we have not pursued this avenue further: Even if a test image and a training image share the same segmentation mask (e.g., a moustache to identify a cat), we cannot assume that the model has correctly learned that cats have moustaches, though this would be an ideal and robust feature to capture.
>
> **Minor points:**
> We have addressed your minor points regarding notation and typos. However, we were unable to find a better alternative name, even though we agree that the current name might be misunderstood as you pointed out.
>
> [1] Julius Adebayo, Justin Gilmer, Michael Muelly, Ian Goodfellow, Moritz Hardt, and Been Kim. Sanity checks for saliency maps. Advances in neural information processing systems, 2018
>
> [2] Kazuaki Hanawa, Sho Yokoi, Satoshi Hara, and Kentaro Inui. Evaluation of similarity-based explanations. In International Conference on Learning Representations, 2021
>
> [3] Fel, T., Picard, A., Bethune, L., Boissin, T., Vigouroux, D., Colin, J., ... & Serre, T. (2023). Craft: Concept recursive activation factorization for explainability. In Proceedings of the IEEE/CVF Conference on Computer Vision and Pattern Recognition (pp. 2711-2721).

---

> > ### Comment · Area_Chair_A4fK · 2025-11-28
> >
> > Dear Reviewer,
> >
> > Please make sure you read the authors' response and engage with them in the discussion before the end of the discussion period on **Dec 03 '25 09:00 PM UTC**. This is a hard deadline.
> >
> > Thank you for supporting quality peer review at ICLR.
> >
> > AC

---

### Author Response · Authors · 2025-11-24
**Revised draft to account for reviewers requests**

The ratings appear somewhat harsh, considering that reviewers acknowledge the paper's contribution in presenting new material clearly and without technical flaws. While we agree that individual use cases could potentially be addressed with existing methods (TDA or FA tailored to specific scenarios) our approach offers a more general framework applicable across multiple use cases simultaneously.

From a conceptual standpoint, our work takes a step towards practical methods for highlighting the idea that the patterns learned by neural networks do not emerge spontaneously; rather, they are derived from training examples. Furthermore, we anticipate that, should the paper be accepted to ICLR, the dissemination of the TFA concept to the computer vision community could inspire the development of additional use cases and applications.

We believe that the general concept of TFA for vision models holds significant interest for the ICLR community, beyond its specific implementation using Grad-Cos and SmoothGrad (we have conducted experiments with other TDA/FA pairs as well). Therefore, we kindly ask the reviewers to reconsider their scores if they feel that our responses below, along with the additional material included in our revised submission, adequately address their concerns.

Compared to the initial draft, we have made the following additions in the revised pdf:

 1. An additional experiment (Appendix B) involving applying FA directly for predicting the training example class for influential examples, as suggested by reviewer AcPD.
 2. An extra quantitative evaluation (Table 1) of our method using deletion metrics, in response to reviewer waGT. We have also moderated our claims to better acknowledge the inherent complexity of evaluating TDA and FA methods.
 3. A more comprehensive ablation study (Appendix C) examining the use of influence functions as the TDA method, alongside raw gradients and Integrated Gradients as FA methods.

We sincerely thank the reviewers for their time and thoughtful consideration.

---

### Meta-Review · Area_Chair_kvLR · 2026-01-02

**Summary:**

The reviewers praised the clarity and good flow of the content of the paper as well as the simplicity of the proposed method. Likewise, the ablation analysis and notes on complexity/computational costs added during the rebuttal were well received.

On the down side, a share concern among all the reviewers was the very limited quantitative evaluation (only based on an insertion experiment) of the proposed method and its proper positioning w.r.t. existing comparable methods. While a deletion experiment was added during the rebuttal, this is nothing more that part of the original insertion-deletion protocol (Petsiuk, 2018) and as such, should have been included from the very beginning. Even then, the quantitative evaluation of the proposed method remains suboptimal.

Beyond that, I agree with Reviewer AcPD on that it is critical to indicate which are the scenarios under which the application of the proposed method offers a clear advantage over existing counterparts. Similarly, I agree with the issue raised by Reviewer E8eB on the confusion that the current title of the paper could cause to the reader.

Vitali Petsiuk, Abir Das, Kate Saenko. “RISE: Randomized Input Sampling for Explanation of Black-box Models” BMVC 2018.

**Reviewer Concerns:**

Addressed Concerns:

- Reviewer E8eB

    - N.A.

- Reviewer Mhg7

    - Missing ablation of the components that constitute the proposed method.

    - Details regarding the number of images used in the spurious correlation experiments.

- Reviewer waGT

    - Missing ablation of the components that constitute the proposed method.

    - Missing details/discussion on the scaling/compute complexity of the proposed method.

- Reviewer AcPD

    - N.A.

Outstanding Concerns:

- Reviewer E8eB

    - Limited quantitative evaluation: only an insertion experiment was considered.

    - Limited comparison w.r.t. a very related work: prototype-based networks.

- Reviewer Mhg7

    - Reduced novelty.

    - Positioning plus empirical comparison w.r.t. existing methods.

- Reviewer waGT

    - Limited novelty, primarly a composition of existing components.

- Reviewer AcPD

    - Missing comparisons w.r.t. simple baselines/variants of the proposed combination of components.

    - Comparison of the proposed method w.r.t. existing methods from the literature.

    - Limited quantitative evaluation.

**Reviewer Scores:**

The authors did an effort to address some of the concerns raised during the rebuttal. Concerns regarding positioning of the proposed method w.r.t to existing efforts and a proper quantitative evaluation remained lightly touched upon. In light of this, I struggle to see how the reviewers would had increased their scores.

---

### Decision · Program_Chairs · 2026-01-26

Reject